# Child Internalizing Problems in Ukraine: The Role of Prosocial and Antisocial Friends and Generalized Self-Efficacy

Viktor Burlaka [1], Oleksii Serdiuk [2,*], Jun Sung Hong [1,3], Lisa A. O'Donnell [1], Serhii Maksymenko [4], Vitalii Panok [5], Heorhii Danylenko [6], Igor Linskiy [7], Valerii Sokurenko [8], Iuliia Churakova [9] and Nadiya Ilchyshyn [2]

1. School of Social Work, Wayne State University, Detroit, MI 48202, USA
2. Research Lab for Psychological Support of Law Enforcement, Kharkiv National University of Internal Affairs, 61080 Kharkiv, Ukraine
3. Department of Social Welfare, Ewha Womans University, Seoul 03760, Korea
4. G.S. Kostyuk Institute of Psychology, National Academy of Educational Sciences of Ukraine, 01033 Kyiv, Ukraine
5. Ukrainian Scientific and Methodical Center of Applied Psychology and Social Work, National Academy of Educational Sciences of Ukraine, 03045 Kyiv, Ukraine
6. Institute for Children and Adolescents Health Care, National Academy of Medical Sciences of Ukraine, 61153 Kharkiv, Ukraine
7. Institute of Neurology, Psychiatry and Narcology, National Academy of Medical Sciences of Ukraine, 61068 Kharkiv, Ukraine
8. Kharkiv National University of Internal Affairs, Ministry of Internal Affairs of Ukraine, 61080 Kharkiv, Ukraine
9. Bloomberg School of Public Health, John Hopkins University, Baltimore, MD 21218, USA
* Correspondence: serdyuk@univd.edu.ua

**Abstract:** The current study examines the association between peer behaviors, self-efficacy, and internalizing symptoms in a sample of 1545 children aged 11 to 13 years old who attended middle schools in eastern Ukraine. We used structural equation modeling (SEM) to examine the role of self-efficacy in the relationship between child internalizing behaviors (anxiety, depression, and somatic complaints) and exposure to prosocial and antisocial friends among girls and boys. Higher self-efficacy was linked with fewer internalizing symptoms for girls and boys. For both boys and girls, exposure to prosocial friends was not statistically associated with changes in internalizing behaviors. However, girls and boys who reported having more antisocial friends had significantly more internalizing symptoms. For girls, association with a greater number of prosocial friends and fewer antisocial friends has been linked with higher self-efficacy and fewer internalizing symptoms. For boys, having more prosocial friends was also linked with higher self-efficacy and fewer internalizing symptoms; however, there was no statistically significant association between having more antisocial friends and self-efficacy. The study discusses the cultural and gender aspects of child socialization in the context of antisocial and prosocial friends, and the development of internalizing behavior problems.

**Keywords:** internalizing behavior problems; generalized self-efficacy; antisocial friends; prosocial friends; anxiety; depression; somatic complaints

## 1. Introduction

Ukraine is the second largest country in Europe with a population of 41.2 million inhabitants [1]. Because of its attractive geopolitical location and unique natural resources, throughout history, generations of Ukrainians have experienced oppression, wars, persecutions, subjugations, and political turmoil [2]. In 2014, in the aftermath of the Revolution of Dignity, Russia invaded and annexed Crimea [3], and supported and recognized separatist parts of the Donbas region. On February 24, 2022, the Russian Federation invaded Ukraine, causing an enormous humanitarian crisis [4]. Given that millions of Ukrainian children

have witnessed the unprecedented devastation, destruction, and violence that may lead to the development of anxiety, depression, and other internalizing symptoms, it is critical to understand the prevalence and the correlates of mental health problems experienced by the Ukrainian youths affected by the military conflict. To our knowledge, the present study uses the largest and the latest data on the mental health of Ukrainian children.

Children in the early adolescence stage (ages 10–14) experience a broad range of developmental changes that might result in internalizing symptoms, which continue to be a major public health problem due to their high prevalence and impairment of functioning. It is estimated that globally, 20.5% of adolescents report experiencing anxiety, and 25.2% show clinically elevated depressive symptoms [5]. Internalizing symptoms are highly correlated with adolescents' decreased academic, social, and interpersonal functioning [6,7], and over the years, a substantial body of research has explored the etiology of adolescent internalizing symptoms, particularly within individual and familial contexts. Several studies have found that adolescents' internalizing symptoms often occur as a result of interaction between genetic and familial risk factors and cognitive emotion regulation strategies [8–10]. Increasing differentiation from parents and greater importance placed on peer relationships are salient markers of the adolescent developmental stage [11], and peer groups hold a significant influence on adolescent behavior and emotional well-being [12,13]. Moreover, according to the social learning theory, adolescents learn about relationships through their interactions with their peer groups [14,15].

Over the years, research has explored whether adolescents' internalizing symptoms are a by-product of the quality of their relations with their peers. Prior research has found a strong association between internalizing symptoms and deviant peer affiliation in adolescents [16,17], which refers to affiliation with peers who are involved in misbehaviors, including stealing, fighting, and drug use [18]. For instance, Brendgen et al. [19], whose study investigated whether friendship with deviant peers would be negatively associated with adolescents' emotional adjustment, found that adolescents with deviant friends showed similar problematic levels of depressive symptoms as their peers without deviant friends. Fergusson et al.'s [16] study, which gathered data from two longitudinal studies in New Zealand, also concluded that adolescents with higher levels of deviant peer affiliations showed a significant increase in depressive symptoms. Other researchers have also documented the protective role of prosocial peers, including, for example, positive qualities in friends [20,21] and secure attachment with peers [22,23] in buffering adolescents' internalizing symptoms. Because adolescents spend a significant amount of time with their friends and peers, prosocial peers such as close friendships can foster adolescents' sense of self-worth, which is likely to reduce the risk of internalizing symptoms, such as depression and anxiety [24,25]. A recent study with 589 Ukrainian students from ten schools in Southern Ukraine revealed that lower scores on prosocial behaviors were associated with a greater number of symptoms of depression [26].

In addition to relationships with peers, research suggests that adolescents with a reduced sense of self and those who have experienced social rejection and social anxiety are more susceptible to deviant peer influences through the peer contagion process [27]. Dishion and Tipsord [27] refer to the peer contagion process as a mutual influence process, which occurs between the individual and peer groups and includes emotions and behaviors, such as aggressive behavior, which may undermine one's development or harm others. However, a similar process can also account for the spread of positive behaviors and emotions among adolescents. For example, both girls and boys who demonstrated prosocial behaviors in Ukraine were likely to have friends that were perceived as kind and helpful [28].

Self-efficacy, another protective factor and a central component of Bandura's [29], social cognitive theory, refers to belief about one's ability to achieve desired goals and actions despite unfavorable life circumstances. Self-efficacy, according to Bandura [29], plays an important role in the self-regulation of one's affective states. In other words, when an individual perceives themselves as ineffective in gaining their desired outcome,

they are likely to become depressed, and when an individual perceives themselves as ill-equipped to deal with potentially threatening events, they are likely to develop feelings of anxiety [30]. Research findings support a strong association between perceived self-efficacy and internalizing symptoms. Findings from Muris' [30] study showed that adolescents with a low level of social self-efficacy displayed social phobia, whereas those with a low level of academic self-efficacy reported school phobia, and those with a low level of emotional self-efficacy showed symptoms of generalized anxiety and panic. In a study on high school students in Iran, Tahmassian and Moghadam [31] also reported a negative association between self-efficacy and anxiety, worried thinking, and social avoidance. Moreover, some studies found a significant mediating role of individual self-efficacy in the association between life adversities and depressive symptoms [32,33]. Forgeard and Benson [34] found that students involved in prosocial, extracurricular activities had a higher sense of mastery and self-efficacy which were in turn related to lower depression and anxiety, and higher psychological wellbeing. These findings provide evidence of the influence of self-efficacy on adolescents' internalizing symptoms.

Currently, no known research has been conducted to examine the relationships between peer behaviors, self-efficacy, and internalizing behaviors. Considering the significant roles peers play in adolescents' social and emotional well-being, peers could potentially influence the development of adolescents' self-efficacy and mental health [27]. However, some children with a stronger sense of self-efficacy may also have the ability to resist negative peer influences, which would likely be negatively related to the development of internalizing symptoms [35]. Individuals who score high on self-efficacy are likely to effectively cope with stressors, such as negative peer influences, as studies have shown that adolescents with high self-efficacy were less likely to engage in risky activities and experience fewer internalizing symptoms [36]. Thus, a higher sense of self-efficacy is critical in protecting adolescents who are vulnerable to negative peer influences in counteracting negative emotions (e.g., depressive symptoms, anxiety) [36,37].

Ukraine is one of the largest countries in Europe, however, there are very few studies of internalizing behaviors among Ukrainian children published in the global peer-reviewed literature. Previous studies have examined the link between drinking, peers, and early onset of alcohol disorder [12] and the association between parenting and internalizing behaviors [8] among Ukrainian children. Applying the social learning theory, the current study examines the association between peer behaviors, self-efficacy, and internalizing symptoms in a large sample of Ukrainian children. We focus on internalizing symptoms, a broader construct of problem behaviors existing within self or directed inwardly (e.g., anxiety, withdrawal, depression, somatic complaints) [38,39]. More specifically, we hypothesize that (1) children involved with more prosocial peers and fewer antisocial peers will have fewer internalizing symptoms, (2) self-efficacy will amplify the protective effect of association with prosocial peers on internalizing symptoms, and (3) children who have more asocial peers will have lower self-efficacy and more internalizing behavior symptoms.

## 2. Method

### 2.1. Participants

The sample included 1545 children attending middle schools in eastern Ukraine. The children attended sixth and seventh grades. The majority of participants were girls (58%). Three-fourths of children (64%) lived in urban and one-fourth lived in rural neighborhoods. Participants' age ranged from 11 to 13 years and the average age was 11.85 years (*SD* = 0.67).

### 2.2. Procedure

The data were collected in 2020–2021 as part of Wave 1 of the Ukrainian Longitudinal Study [40]. All procedures were approved by the Commission on Ethics and Deontology of the Institute of Neurology, Psychiatry and Narcology of the National Academy of Medical Sciences of Ukraine (Protocol No 12-b from 21 December 2018) and the University of Michigan Medical School Institutional Review Board IRBMED (Study eResearch ID:

HUM00156825; OHRP IRB Registration Number: IRB00000244 from 18 January 2019). The participants were informed of the study's aim, which was to monitor the impact of various factors on the health of Ukrainian children throughout life with a special focus on risky behaviors (mostly addictive behavior). Parents signed the informed consent, and children signed informed assent. Data were collected by research assistants using a web-based questionnaire distributed through a secure online platform. Children answered questions using individual school or personal computers, tablets, or smartphones in school IT classrooms or at home. Each child received a personal link to the questionnaire with an individual participant code. Children were informed that their depersonalized data were stored in a secure data set and only the team of researchers had access to the children's answers. Children spent approximately one hour answering questions about personal experiences, school, and family, mental and physical health, and substance use.

### 2.3. Measures

Participants answered the demographic questions on sex at birth (male, female) and age (measured in years).

The Youth Self-Report [38] was used to assess internalizing symptoms. This measure has been professionally translated and back-translated from Ukrainian to English. The YSR is a well-known measure that has been previously used to measure internalizing symptoms of Ukrainian children [8]. The answers to YSR problem items range from *not true (0)*, *somewhat or sometimes true (1)*, and *very true or often true (2)*. The YSR internalizing scale measures children's anxiety, depression, somatic, and withdrawn symptoms (e.g., "I am afraid of going to school", "I am self-conscious or easily embarrassed", "I feel worthless or inferior", "I would rather be alone than with others", "I keep from getting involved with others", "Physical problems without known medical cause: aches, pains"). Cronbach $\alpha$ for this study was 0.91.

The Peer Behavior Profile (PBP) [41] was used to assess peer involvement in prosocial behaviors (Cronbach $\alpha$ = 0.66) and antisocial behaviors (Cronbach $\alpha$ = 0.81). The PBP has been previously used in studies of adolescent internalizing symptoms [42] prosocial scale Cronbach $\alpha$ = 0.89, antisocial scale Cronbach $\alpha$ = 0.93). The 23-item instrument uses a 5-point Likert scale to assess the proportion of friends who engage in certain behaviors ranging from *almost none (1)* to *nearly all (5)*.

Self-efficacy was assessed with the Generalized Self-Efficacy Scale (GSE) [43]. The 10-item GSE scale can be used with adolescents and adults and taps into a construct of optimistic self-belief and positive resistance. The response ranges from *not at all true (1)* to *exactly true (4)*. The reported Cronbach $\alpha$ for GSE ranged from 0.76 to 0.90 [43]. In the current study, the GSE Cronbach $\alpha$ was 0.91.

### 2.4. Analytic Techniques

Structural equation modeling (SEM) (see Figure 1) was used to simultaneously assess direct effects and indirect effects for girls and boys. SEM is a well-established technique that separates the random measurement error from latent variables to increase the explanatory power. SEM allows estimating direct and indirect effects that are "routinely included in structural models, assuming such specifications are theoretically justifiable" [44]. Kline's [44] recommendations were followed in this study to evaluate SEM identification and maximum likelihood estimation was used to examine the model fit using Stata/MP 16.1 software package [45]. To examine the goodness-of-fit we used the comparative fit index (CFI), Tucker–Lewis index (TLI), and the root means squared error of approximation (RMSEA) [46]. The cutoff point for an acceptable fit for CFI and TLI was 0.95 or above and for RMSEA, 0.08 or below [47]. Stata's delta method-based nlcom command was used to examine the indirect relationships in the model [45,47,48]. The tests for group invariance of parameters were conducted in Stata for the groups of girls and boys. Stata estimates both groups simultaneously without equality constraints across the two groups. We used

the $\chi^2$ (Wald) tests ('estat ginvariant' command) to examine which parameters differed significantly between boys and girls [47].

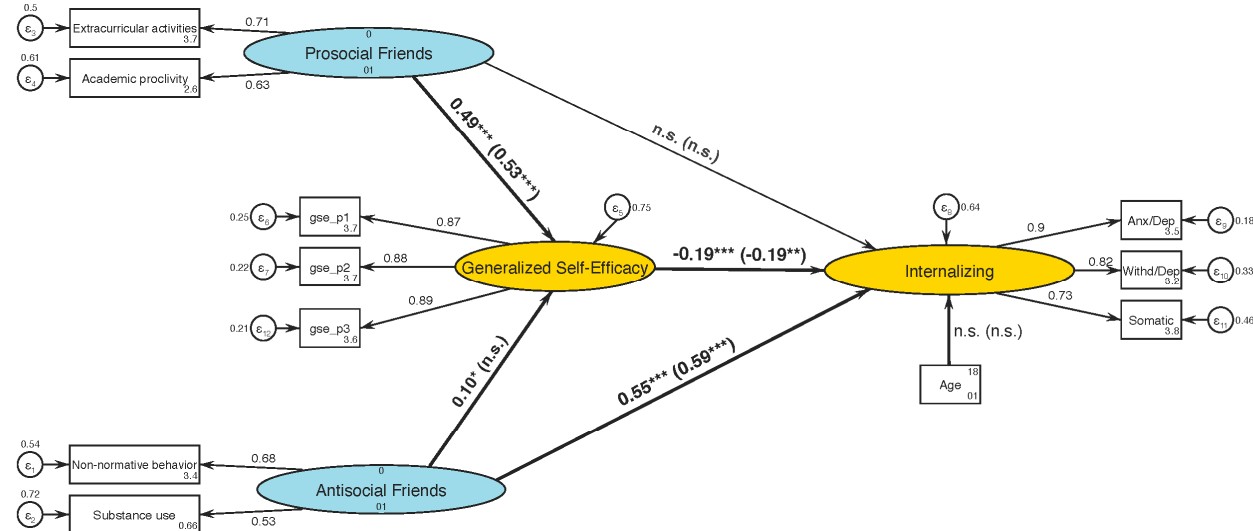

**Figure 1.** Structural Equation Model. Final structural equation model representing effects of prosocial and antisocial friends and generalized self-efficacy on past 6 months youth self-report of internalizing problems among 1545 Ukrainian schoolchildren aged 11–13 years. Ellipses represent latent constructs. The paths are shown as straight lines and the values along the lines are standardized path coefficients significant at n.s. = "non-significant", * $p < 0.05$, ** $p < 0.01$, *** $p < 0.001$ for girls (first values) and boys (parenthesized values).

## 3. Results

Our findings indicate that the mean YSR internalizing score was 1.39 (SD = 0.32) in this sample of Ukrainian children. A total of 32% of girls and 25% of boys answered that they feel too fearful and anxious. A total of 30% of girls and 23% of boys stated that they sometimes felt unhappy, sad, or depressed. Girls were also more likely than boys to experience somatic symptoms, such as stomachaches (39% of girls and 31% of boys), or headaches (55% of girls and 42% of boys). Children reported that 27% of their friends used tobacco, 28% used alcohol, and 3% used illegal drugs. Table 1 provides additional information on study variables and significant correlations.

**Table 1.** Means, standard deviations, ranges, and intercorrelations among study variables (N = 1545).

|  | % | M | SD | Range | 1 | 2 | 3 | 4 | 5 | 6 | 7 |
|---|---|---|---|---|---|---|---|---|---|---|---|
| 1. YSR internalizing |  | 1.39 | 0.32 | 1–3 | — |  |  |  |  |  |  |
| 2. Age |  | 11.85 | 0.67 | 11–13 | 0.05 * | — |  |  |  |  |  |
| 3. Sex, girls | 58.45 |  |  |  | −0.12 *** | 0.00 | — |  |  |  |  |
| 4. Non-normative friends |  | 1.46 | 0.46 | 1–5 | 0.33 *** | 0.07 ** | 0.04 | — |  |  |  |
| 5. Friend's substance use |  | 1.96 | 0.29 | 0–1 | 0.28 *** | 0.13 *** | 0.02 | 0.32 *** | — |  |  |
| 6. Friends extracurricular |  | 2.82 | 0.75 | 1–5 | −0.01 | 0.05 * | 0.01 | 0.23 *** | 0.03 | — |  |
| 7. Friends academic proclivity |  | 2.49 | 0.93 | 1–5 | −0.10 *** | 0.02 | −0.06 * | 0.08 *** | −0.11 *** | 0.46 *** | — |
| 8. Generalized self-efficacy |  | 2.69 | 0.68 | 1–4 | −0.21 *** | 0.05 | −0.02 | 0.04 | −0.07 * | 0.34 *** | 0.30 *** |

Note: * $p < 0.05$. ** $p < 0.01$. *** $p < 0.001$.

Results suggest that both girls ($b = -0.19$, $p < 0.001$) and boys ($b = -0.19$, $p < 0.01$) had fewer internalizing symptoms if they had higher self-efficacy.

Girls ($b = 0.55$, $p < 0.001$) and boys ($b = 0.59$, $p < 0.001$) who had more friends with antisocial behaviors were more likely to develop internalizing symptoms. However, having more friends who engage in prosocial behaviors did not have a significant association with internalizing behaviors. For girls, having more ($b = 0.49$, $p < 0.001$) friends with prosocial behaviors and fewer ($b = -0.10$, $p < 0.05$) friends with antisocial behaviors was associated with higher general self-efficacy. For boys, having more prosocial friends ($b = 0.53$, $p < 0.001$)

was linked with higher general self-efficacy scores while the relationship between having antisocial friends and GSE scores was not statistically significant. The paths' coefficients from prosocial friends, antisocial friends, and generalized self-efficacy to internalizing behaviors were not statistically different across the two groups (tests for girls versus boys group parameter differences: $p = 0.894$ for antisocial peers; $p = 0.920$ for prosocial peers; $p = 0.737$ for the self-efficacy).

The higher number of prosocial friends had a significant standardized indirect association with lower internalizing symptoms, which was mediated by the generalized self-efficacy, both for girls ($b = -0.10$, $p < 0.001$) and for boys ($b = -0.10$, $p < 0.001$). Additionally, having more friends with antisocial behaviors had a significant standardized indirect association, mediated by generalized self-efficacy, with higher internalizing symptoms among girls ($b = 0.02$, $p < 0.05$), but not for boys. The model provided a good fit for the data: $\chi^2$ (92, $N = 1542$) = 362.46, $p < 0.001$, CFI = 0.96, TLI = 0.95, RMSEA = 0.062.

## 4. Discussion

There have been fewer studies on internalizing behaviors because such problems as depression, anxiety, and somatic complaints are less visible than classroom disruptive behaviors, substance abuse, or aggression [49]. Researchers often have limited access to children of school age while teachers are less likely to notice internalizing problems and sometimes may even misinterpret them as good behaviors. Furthermore, the measurement of internalizing problems is often problematic because the teacher and parent reports on child internalizing behaviors often rely on guesses about children's inner feelings rather than more readily observable externalizing behaviors such as thefts or acts of violence. We used self-reported data from a large sample of Ukrainian children collected during the Russian invasion of Ukraine.

A significant proportion of Ukrainian children reported internalizing behavior problems at much higher rates than those previously reported in the global literature [5]. Since internalizing symptoms are highly correlated with impairments in adolescents' functioning [6], it is critical to use research to establish a baseline of mental health problems among Ukrainian children affected by the military conflict to be able to examine the cultural aspects of coping with adversity. It is also important from the standpoint of using future research to document any changes in the mental health status to be able to understand the significance of the war for the Ukrainian children and develop culturally sensitive interventions to mitigate such symptoms. This study explored the association between peer behaviors, self-efficacy, and internalizing symptoms in adolescents, as peers exert tremendous influence on adolescent behavior and emotional wellbeing [13].

Our results suggest antisocial and prosocial peers have differential patterns of association with internalizing problems. The findings indicate that involvement with antisocial peers had a significant direct effect on internalizing symptoms both for girls and boys. One of the possible explanations for this finding could be that participants could share life circumstances with their antisocial peers, such as early adversity and difficulties in relationships with parents as well as having a higher risk of engaging in rule-breaking, sexual risk-taking, substance abuse with negative consequences that could lead to depression [16]. Another plausible pathway to the development of internalizing behaviors includes problem parent-child relationships, lack of social skills, increased risk of peer victimization, and subsequent affiliation with deviant peers, a process that often co-occurs with externalizing and internalizing problems [18].

The relationship between prosocial peer affiliation and internalizing problems was not statistically significant in our sample. This finding suggests that having more friends who pursue such prosocial goals as academic excellence and engagement in extracurricular activities does not necessarily protect children from the development of anxiety and depression. Previous research examined the role of support from close friends on a child's mental health. Indeed, higher quality of friendships has been associated with lower social anxiety among early adolescents [24]. La Greca and Harrison [20] also found that social anxiety

was negatively associated with support from close friends. Likewise, Allen et al. [22] found that the overall quality of peer relationships was linked with attachment security, and that attachment security was associated with lower depression among adolescents. In our study, however, children were asked about the behaviors of all friends, not only the 'close friends.' Because our analyses utilized variances reflecting the association with both close friends and acquaintances/social media friends, it may have diluted the strength of the link between association with prosocial friends and child mental health. Future studies should take into account the degree of emotional closeness and attachment aspects as a way to examine the relationship between internalizing behaviors and association with prosocial peers.

Rather than focusing on the quality of peer relationships, our study has explored the possibility that exposure to prosocial peers can have indirect effects on a child's mental health, for example, through learning and imitation of prosocial skills that can enhance children's resilience to adverse life circumstances. Our findings have been consistent with prior research in that self-efficacy is a protective mechanism that shields adolescents from developing anxiety and depression [30,31]. Ukrainian children who had more skills to withstand difficult life situations were less likely to develop internalizing symptoms resulting from various life adversities [32,33]. Furthermore, the study found that both girls and boys who had more friends with prosocial behaviors were more likely to acquire self-efficacy skills and experienced fewer symptoms of depression, anxiety, and somatic complaints.

A different pattern was observed concerning the role of antisocial friends. Specifically, having more antisocial friends has been linked with a lowered sense of self-efficacy among girls but not boys. This result can be best interpreted in the context of Ukrainian culture and expectancies regarding gender socialization. Świetlicki [50] analyzed gender roles in Soviet and Ukrainian books for girls and boys. The author noted that girls and women were often offered advice on beauty, housekeeping, and taking steps to attract males and eventually marry them. Women were expected to be clean and refined. In this sample, some girls may have engaged in substance abuse and sexual risk-taking with antisocial friends and experienced victimization. Given cultural expectations, such experiences could bring about a sense of losing one's purity and wholeness. Life traumas may have led to some girls feeling broken and defective in the face of societal expectations of purity, healthfulness, and beauty. These girls may have stopped believing that they can successfully deal with life adversity and developed internalizing symptoms.

In contrast, according to Świetlicki [50], Ukrainian boys acquire, through books, a sense of personal agency and the fluid ability to respond to life adversities. For example, the books told Ukrainian boys that almost everything has been designed by men and that men must fight back and protect themselves. Boys are not expected to be pure and beautiful; they are often not judged for making mistakes, which makes it easier for the boys to make a comeback after experiencing adverse, and potentially embarrassing life events. Our results suggest that some Ukrainian boys can get involved with antisocial peers while still believing that they are somewhat independent in their ability to deal with stress, that a failure does not necessarily mean a lost cause, and that their life can sooner or later take a turn for the better. These results warrant future qualitative research to verify these assumptions and get a deeper understanding of gender differences in coping strategies among Ukrainian children.

It is important to interpret these findings in light of the limitations, such as cross-sectional research design, the reliance on self-reports exclusively, and data collection in urban settings. Although these preliminary results provide important evidence on the role of self-efficacy in explicating the association between peer behaviors and the development of internalizing symptoms, future research needs to replicate these findings using longitudinal and experimental designs and with multi-informants (e.g., parents, peers, and teachers). Furthermore, the data for the present study were predominantly collected in eastern Ukraine, and the sample consisted of children attending sixth and seventh grades. It is not clear how these findings generalize to children of other age groups and those living

in western regions of Ukraine that have been affected by the Russian invasion of Ukraine to a lesser degree. In addition, future research should use qualitative approaches to examine the nature of gender differences in self-efficacy skills used by Ukrainian children.

Additional information is needed on the risk factors for internalizing symptoms among children, especially within peer contexts. It is also important to continue testing ways in which friends and peers are influential in the development of internalizing symptoms among Ukrainian children. Specifically, future studies might also explore whether adolescents without friends are at risk of similar internalizing symptoms as those with antisocial peers [19]. Most importantly, findings from the current and future studies need to inform culturally relevant practice for Ukrainian children showing the signs of depression, anxiety, and other internalizing symptoms.

## 5. Conclusions

Taken together, our results suggest that peers' prosocial behaviors have no direct association with children internalizing problems. However, socializing under the influence of antisocial peers may increase the risk of developing internalizing symptoms for boys and girls. Additionally, having more prosocial and fewer antisocial friends is associated with a significant increase in self-efficacy and a subsequent reduction in internalizing symptoms for girls. For boys, having more prosocial friends is also linked with higher self-efficacy and fewer internalizing symptoms.

**Author Contributions:** Conceptualization, V.B. and O.S.; methodology, V.B. and O.S.; formal analysis, V.B. and O.S.; investigation, V.B., O.S., S.M., V.P., H.D., I.L., V.S. and N.I.; data curation, V.B. and O.S.; writing—original draft preparation, V.B., O.S., J.S.H., L.A.O., S.M., V.P., H.D., I.L., V.S., I.C. and N.I.; writing—review and editing, V.B., O.S., J.S.H., L.A.O., S.M., V.P., H.D., I.L., V.S., I.C. and N.I.; visualization, V.B.; project administration, O.S. All authors have read and agreed to the published version of the manuscript.

**Funding:** This study was supported in part by the National Institutes of Health/Fogarty International Center Capacity Building for Lifespan Focused Substance Use Disorder Research in Ukraine Grant (4D43TW009310-05).

**Institutional Review Board Statement:** All procedures were approved by the Commission on Ethics and Deontology of the Institute of Neurology, Psychiatry and Narcology of the National Academy of Medical Sciences of Ukraine (Protocol No 12-b from 21 December 2018) and the University of Michigan Medical School Institutional Review Board IRBMED (Study eResearch ID: HUM00156825; OHRP IRB Registration Number: IRB00000244 from 18 January 2019).

**Informed Consent Statement:** Informed consent was obtained from all subjects involved in the study.

**Data Availability Statement:** The data presented in this study are available on request from the corresponding author. The data are not publicly available due to privacy restrictions.

**Conflicts of Interest:** The authors declare no conflict of interest.

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
