# Peer review of "Child Internalizing Problems in Ukraine: The Role of Prosocial and Antisocial Friends and Generalized Self-Efficacy"

_societies, doi:10.3390/soc12050144_

Round 1

Reviewer 1 Report

Thank you for the opportunity to review the manuscript entitled, "Child internalizing problems in Ukraine: The role of prosocial and antisocial friends and generalized self-efficacy."  This manuscript investigates relationships between peer relationships, self-efficacy, and internalizing problems for middle school students in Ukraine, in the context of the Russian invasion.  As such, it comprises a unique and important contribution to the literature.  Points of feedback are below.

Most substantively, I have questions pertaining to the measurement of prosocial and antisocial peers.  Is it possible that some peers have both prosocial and antisocial attributes?  Is it possible that some children have both prosocial and antisocial peers?  I assume that the answers to these questions are yes - and I wonder how that might impact the results of the study.

It is not clear to me whether bidirectional influences between self-efficacy and friendships were tested.  Wouldn't self efficacy also lead to the development of more friendships (in addition to friendships fostering self-efficacy, as the authors describe)?

Also, the timing of data collection with regard to the Russian invasion is unclear.  In the discussion, the authors note that data was collected during the invasion.  In the methods, the authors say that data collection stopped due to the Russian invasion.  

On line 62, I think you meant to insert the word 'deviant' before the word 'friends'.

On line 81, there is a closed parentheses that is extraneous.

In the discussion, while I agree that it might be more difficult to accurately measure internalizing than externalizing behaviors, I don't think it is necessary to say that there are fewer studies on internalizing behaviors.  There is quite a large literature base; the quantity of studies does not detract from the contributions of this manuscript.

Author Response

Response to Reviewer 1 Comments

Point 1: Most substantively, I have questions pertaining to the measurement of prosocial and antisocial peers.  Is it possible that some peers have both prosocial and antisocial attributes?  Is it possible that some children have both prosocial and antisocial peers?  I assume that the answers to these questions are yes - and I wonder how that might impact the results of the study.

Response 1:

We agree with the reviewer that children are influenced by prosocial and antisocial peers. Therefore, both “Generalized Self-Efficacy” and “Internalizing” latent variables are represented by the overlapping variance of “Prosocial Friends” and “Antisocial Friends.” The advantage of using the SEM analytic approach over other approaches is that it examines the positive and negating peer influences simultaneously, rather than separately, concerning generalized self-efficacy and internalizing behaviors.

Point 2: It is not clear to me whether bidirectional influences between self-efficacy and friendships were tested. Wouldn't self efficacy also lead to the development of more friendships (in addition to friendships fostering self-efficacy, as the authors describe)?

Response 2: We thank the reviewer for their thoughtful comment. The choice of the model in this study is primarily based on theory and previous research elsewhere on the increased risks for children to develop mental health problems due to deviant peer affiliation. The novelty of this study is that it a) tested this association for the first time with Ukrainian children, and b) recognized the role of the innate characteristics (self-efficacy) of the child that can potentially mitigate the negative peer influences (in essence, looking at resilience to the given adversity that surrounds the child rather than examining the child’s ability to create friendships), and c) explored differences in these behaviors among boys and girls. We also tested the alternative model, as suggested by the reviewer, in which prosocial and antisocial peers mediated the relationship between GSE and internalizing behaviors but it provided a poorer fit than the model we use in our study.

Point 3: Also, the timing of data collection with regard to the Russian invasion is unclear. In the discussion, the authors note that data was collected during the invasion. In the methods, the authors say that data collection stopped due to the Russian invasion.

Response 3: Thank you for your feedback. We agree that it sounds confusing that data were collected during the invasion and stopped due to the invasion. This is because there have been two waves of invasion—a smaller one that affected parts of Ukraine and a full-scale wave that affected the entire country. The first wave began in 2014 when the Russian military took Crimea and established a presence in two regions in Eastern Ukraine. We collected data in close proximity to occupied territories. Some participants could be worried that fighting was happening close to their homes but most have not directly witnessed the conflict themselves. In February 2022, the situation became worse in that Russian troops were all over the country and the fighting came to participants’ neighborhoods and homes. The entire territory of Ukraine has been engulfed in war. There has been extensive shelling and substantial destruction of infrastructure (one missile hit and destroyed a building where one of the study investigators used to work). Millions of families became either internally displaced or refugees. Given the uncertainty and direct risk to life, the study was suspended and at the time of manuscript preparation we wrote that the data collection was “stopped”. However, after careful consideration, the research team has decided to continue the study to examine how witnessing the war has affected our participants. We thank the reviewer for this thoughtful comment and removed the language about study termination from the manuscript.

Point 4: On line 62, I think you meant to insert the word 'deviant' before the word 'friends'.

Response 4: Thank you, we have added the word deviant in the revised version.

Point 5: On line 81, there is a closed parentheses that is extraneous.

Response 5: Thank you, we deleted it.

Point 6: In the discussion, while I agree that it might be more difficult to accurately measure internalizing than externalizing behaviors, I don't think it is necessary to say that there are fewer studies on internalizing behaviors.  There is quite a large literature base; the quantity of studies does not detract from the contributions of this manuscript.

Response 6: Thank you for this feedback. The reviewer is right that there is a substantial literature base on internalizing behaviors. The reason for the inclusion of this paragraph was the attempt to draw attention to often overlooked children who might sit back in the corner of the classroom and can be perceived as “well-behaving” by schoolteachers when in fact, they are going through terrible experiences of feeling lonely, empty, and scared. As discussed by Cicchetti & Toth (2014) this is a universal problem that aggression and some of the more “visible”, disruptive behaviors often get more attention from helping systems and researchers. Our study shows that it is both feasible and important to conduct research that gives voice to people who suffer from symptoms of depression and anxiety.

Reviewer 2 Report

The paper addresses an important topic, is well written, logical in its arguments, and presents an interesting analysis.

One comment is that it would be worth discussing briefly how representative you think the sample is of the underlying population; and whether there is any possibility of sample selection bias that might be influential here.

Author Response

Response to Reviewer 2 Comments

Point 1: One comment is that it would be worth discussing briefly how representative you think the sample is of the underlying population; and whether there is any possibility of sample selection bias that might be influential here.

Response 1: We appreciate the reviewer’s helpful comment and have updated the discussion in the study limitations section (please see lines 334-337): “[…] the data for the present study were predominantly collected in the Eastern Ukraine, and the sample consisted of children attending 6th and 7th grades. It is not clear how these findings generalize to children of other age groups and those living in Western regions of Ukraine that have been affected by the Russian invasion of Ukraine to a lesser degree.”
